# G4Beacon: An In Vivo G4 Prediction Method Using Chromatin and Sequence Information

**DOI:** 10.3390/biom13020292

**Published:** 2023-02-03

**Authors:** Zhuofan Zhang, Rongxin Zhang, Ke Xiao, Xiao Sun

**Affiliations:** State Key Laboratory of Bioelectronics, School of Biological Science and Medical Engineering, Southeast University, Nanjing 210096, China

**Keywords:** G-quadruplex, in vivo G-quadruplex prediction, Gradient-Boosting Decision Tree (GBDT), chromatin accessibility

## Abstract

G-quadruplex (G4) structures are critical epigenetic regulatory elements, which usually form in guanine-rich regions in DNA. However, predicting the formation of G4 structures within living cells remains a challenge. Here, we present an ultra-robust machine learning method, G4Beacon, which utilizes the Gradient-Boosting Decision Tree (GBDT) algorithm, coupled with the ATAC-seq data and the surrounding sequences of in vitro G4s, to accurately predict the formation ability of these in vitro G4s in different cell types. As a result, our model achieved excellent performance even when the test set was extremely skewed. Besides this, G4Beacon can also identify the in vivo G4s of other cell lines precisely with the model built on a special cell line, regardless of the experimental techniques or platforms. Altogether, G4Beacon is an accurate, reliable, and easy-to-use method for the prediction of in vivo G4s of various cell lines.

## 1. Introduction

A G-quadruplex (G4) is a non-canonical structure that forms in guanine-rich sequences [1]. The basic unit of the G4 is the G-quartet, a planar structure of four Hoogsteen hydrogen-bounded guanines with a cation such as potassium in the middle of the plane, which can stabilize the structure [2,3]. Guanine-rich sequences with special patterns can form two or more G-quartets and stack them into diverse topologies, including parallel and antiparallel modes [2]. G-quadruplexes have been reported to be enriched in some special regions, including promoters, telomeres, and double-strand breaks (DSBs), etc. [4,5], which are tightly linked to their functions.

The G4-detection techniques can be classified into two categories: experimental methods and computational methods. Early experimental techniques of G4-detection, such as NMR [6] and X-ray crystallography [7], focused on the structure information, while recent methods based on next-generation sequencing (NGS) techniques provide genome-wide information of the sequences and loci of G4s [8]. The most widely used G4 sequencing techniques are G4-seq [9] and G4 ChIP-seq [10]. G4-seq is an in vitro G4 detection method that combines DNA polymerase stalling and NGS techniques to detect the G-quadruplex structures under the non-intracellular environmental state, and the number of G4s detected by G4-seq is approximately 700 thousand in the human genome [9]. Recent research employed a simpler DNA-seq experiment approach and proposed a sequencing quality analysis method to profile whole-genome G4 sites [11]. Different from the G4-seq method, G4 ChIP-seq uses the immunoprecipitation technique to map endogenous G4s in genomes, detecting approximately 10 thousand in vivo G4s (also called active G4s) in human cells such as HaCaT and K562 [10,12]. A recent study also attempted to detect G4s in living cells using an artificial G4 probe protein (G4P), reporting similar amounts of in vivo G4s in different human and non-human cells [13]. Besides the above methods, the Cleavage Under Targets and Tagmentation (CUT&Tag) technique was also used for genome-wide in vivo G4 mapping, which showed better performance in terms of G4 detection sensitivity but suffered from detection failure in fixed tissue compared to other G4 probing methods [14,15,16]. The quantity of G4s identified in vivo is significantly lower than that in vitro, suggesting that the cellular environment may play an important role in G4 production in cells [2,10,12].

Except for the experimental methods discussed above, there are several computational methods (also known as in silico methods) designed to predict G4 sequences. Most of these algorithms are used to detect the potential in vitro G4 sites without considering the specific cellular environment [8]. The first generation of G4 prediction algorithms are regular-expression (regex)-based methods through empirical regexes. For example, the early regex-based algorithm Quadparser uses $G_{3-5}N_{1-7}G_{3-5}N_{1-7}G_{3-5}N_{1-7}G_{3-5}$ and has identified approximately 376,000 G4s in the human hg19 genome [17]. The further developed algorithms are called scoring-based algorithms [8], using more flexible regexes and scoring methods to detect more potential G4s. For instance, the representative scoring-based algorithm QGRS mapper uses $G_{x}N_{y1}G_{x}N_{y2}G_{x}N_{y3}G_{x}\;\left(x\geq 2\right)$ to search for motifs and utilizes G-scores to evaluate whether a candidate motif can form a stable G4 structure [18]. The G-score method tends to give a high score when a motif has (i) relatively short loops, (ii) equal or similar loop sizes, and (iii) a great number of guanine tetrads [18]. After the G4-seq approach was proposed, more non-canonical G4s, such as long-loop and bulged structures, were found [9], which brought more complex sequence patterns for computational methods to detect. To improve the performance in predicting non-canonical G4s, methods using the sliding-window scoring strategy (G4Hunter) [19] or data-driven models, such as Quadron [20] and G4detector [21], were proposed.

Following the development of in vivo G4 detection experiments, an individual computational method for predicting active G4 regions in cells, DeepG4, was presented [22]. DeepG4 is a data-driven method using sequence features appending cellular environment features such as chromatin accessibility information. It employs a convolutional neural network (CNN) and scans whole input regions with a fixed-length (200 bp) window [22]. However, this method suffered from inadequate performance in whole-genome in vivo G4 mapping: the AUPRC and F1-score declined heavily when the model was tested on the whole-genome scale test sets [22]. These results reflect the fundamental challenge of the whole genome in vivo G4 mapping problem, which is called the data-imbalanced problem. The skewed distribution of G4 data brings serious training difficulties and prediction challenges [23]. Most traditional machine learning models, as well as deep learning models, are not suitable for imbalanced dataset training. Therefore, balancing preprocessing methods should be employed before model training. Moreover, more expressive features are also required to improve the performance, as other negative factors such as the existence of noisy samples will be enlarged by imbalanced data problems [23].

In this study, we proposed a cell-type-specific G4 prediction tool called G4Beacon, which implemented the Gradient-Boosting Decision Tree (GBDT) algorithm to predict if a G4-seq entry can fold into the quadruplex structure in a given cell type (Figure 1). The main idea of G4Beacon is to use the ATAC-seq signal data and the surrounding sequences of G4-seq entries that can provide information on chromatin accessibility as well as sequence patterns to learn and identify whether a specific G4 site (G4-seq entry) could form the quadruplex structure in a given cell (Figure 1 and Appendix A). In our model, the cell-type-specific positive samples were defined as those G4-seq entries that well overlapped with in vivo G4s, while the remaining G4-seq entries were considered as negative samples. To avoid the influence of data imbalance on training in G4Beacon, we adopted an oversampling strategy to balance the data ratio. We then inspected the performance of G4Beacon by designing and applying one-cell-line and cross-cell-line experiments. We confirmed that G4Beacon is a robust and accurate method, even when working with datasets that are highly skewed or derived from other G4 probing techniques. In conclusion, G4Beacon can precisely identify the in vivo G4s of previously uncharacterized cell lines in a short time using only a few commonly used features, which can greatly improve the current dilemma of poor in vivo G4 data.

## 2. Materials and Methods

### 2.1. Raw Data

To build and evaluate G4Beacon, the ATAC-seq data and in vivo G4 sequencing data were required for feature construction and dataset division. The critical problem was that there were few in vivo G4 sequencing data due to the high cost and experimental difficulty of in vivo G4 sequencing. We required cell lines with both high-quality in vivo G4 sequencing and ATAC-seq data, and, finally, three human cell lines—K562 (human chronic myelogenous leukemia cell line), HepG2 (human hepatoblastoma cell line), and MCF7 (human breast adenocarcinoma cell line)—were collected and used in this research.

The human G4-seq data were downloaded from the Gene Expression Omnibus (GEO) database with the accession number of GSE110582 [24]. We used G4-seq entries derived from the K+ experimental environment. The G4 ChIP-seq data for K562 and HepG2, and the G4 CUT&Tag data for MCF7 cell lines, were retrieved from the GEO database with the accession numbers GSE107690, GSE145090, and GSE181373, respectively. The cell-type-specific ATAC-seq data were used for the construction of cellular environment features. The ATAC-seq data were obtained from the Encyclopedia of DNA Elements (ENCODE) portal with the identifiers of ENCFF357GNC, ENCFF262URW, and ENCFF976UNK for the cell lines of K562, HepG2, and MCF7, respectively. The histone modification state data were also obtained from the ENCODE database with the identifiers of ENCFF783QIW (K562-H3K4me1), ENCFF060FHP (K562-H3K4me2), ENCFF370CHI (K562-H3K4me3), ENCFF541RTV (K562-H3K9ac), ENCFF422BDQ (K562-H3K9me3), ENCFF465GBD (K562-H3K27ac), ENCFF796REQ (K562-H3K27me3), ENCFF199WYG (MCF7-H3K4me1), ENCFF024VOG (MCF7-H3K4me2), ENCFF615NAU (MCF7-H3K4me3), ENCFF917IBG (MCF7-H3K9ac), ENCFF642EON (MCF7-H3K9me3), ENCFF246QSB (MCF7-H3K27ac), and ENCFF216VVY (MCF7-H3K27me3). The human genome assembly version of hg19 was used as the reference genome. All the epigenetic data that originated from other assembly versions were then converted into hg19 by utilizing the UCSC liftOver tool [25].

### 2.2. Positive/Negative Sample Division

We classified G4-seq entries into two parts, namely active G4 entries (positive samples) and inactive G4 entries (negative samples), for each cell line according to whether these G4-seq entries were supported by in vivo G4 experiments (Figure 2). We used bedtools [26] to overlap the G4-seq entries and the G4 ChIP-seq/G4 CUT&Tag entries, taking the overlap length as the criterion to accept a G4-seq entry as a positive sample or not. Considering that the resolution of G4 ChIP-seq [27] and G4 CUT&Tag experiments was approximately 100–500 bp [16], only the G4-seq entries that overlapped with G4 ChIP-seq/G4 CUT&Tag entries were retained and regarded as positive samples [17,19], while other G4-seq entries were treated as negative samples. We used 10% of the G4 ChIP-seq/G4 CUT&Tag peak length as the overlapping threshold, which was considered to mitigate the false positive problem (Appendix A). After sample division, we finally obtained positive/negative samples for each cell line (see Table 1).

Due to the imbalance of positive and negative samples, which may affect the performance of traditional machine learning algorithms [23], training set balancing is essential before it is used for model training. We compared different data sampling strategies (Appendix A) and then applied an over-sampling approach to ensure an equal size of positive and negative samples (see Section 2.5).

### 2.3. Feature Selection and Construction

G4Beacon took both the chromatin accessibility data (ATAC-seq) and the sequence data to construct features for each input G4-seq entry in the candidate dataset (Figure 1). To acquire more information about the chromatin environment of the surrounding genomic regions, each G4-seq entry was then extended by 1 kb upstream and downstream of its center, respectively—that is, a 2 kbp length region (Figure 1).

The chromatin environment can influence the folding capability of in vivo G4s, as they were found to appear together with ATAC-seq peaks [10]. The ATAC-seq signal value for each G4-seq entry region was calculated as follows. First, the G4-seq entry region was divided into 200 non-overlapping windows with 10 bp resolutions; second, the average ATAC-seq signal value was computed within each 10 bp window. As a result, a vector of imputed chromatin accessibility values with a size of 200 × 1 for each sample was obtained for the subsequent analysis.

Most of the existing G4 prediction models merely use the sequences from G4 sites to learn the sequence patterns of G4 structures [21,22]. However, in our model, the sequence of each candidate G4-seq entry along with its surrounding regions was also extracted. Since it is not an optimal strategy to apply the one-hot encoding method in tree learners, we used the ordinal categorical encoding method for the construction of our model, as inspired by LightGBM [28]. Generally, a raw 2000 × 1 sequence feature vector is taken as input and, for each position of the sequence, the variable in the {A, T, C, G} set is mapped into {0, 1, 2, 3} directly. These 2000 positions in the feature vector will be marked as categorical variables using the categorical variable option of LightGBM (see Section 2.4) [28].

### 2.4. Machine Learning Model: Gradient-Boosting Decision Tree (GBDT)

We compared the performance of several machine learning methods, including Logistic Regression, Decision Tree, Random Forest, and Gradient Boosting Decision Tree (GBDT) (Appendix A), and finally the GBDT algorithm performed better than other models and was used to construct the G4 folding capability prediction model [29]. The GBDT algorithm is a dataset-scale-robust boosting method that is largely applied in academic research as well as in industrial applications. The principal mechanism of gradient boosting is to use multiple base estimators and train each estimator to fit the difference between the former estimator results and the true labels. There are two high-performance and widely used implements of GBDT: Xgboost [30] and LightGBM [28]. In this study, we utilized the Python API of LightGBM version 3.2.1 (https://github.com/microsoft/LightGBM, accessed on 2 November 2022) to implement our model workflow as it can provide categorical feature support and high training efficiency [28].

As mentioned in Section 2.3, we enabled the categorical variable tag for each position in the sequence feature. Instead of using the one-hot encoding method, which may lead to unbalanced growth and high complexity problems, LightGBM sorts the categories according to the training objective at each split and finds the best split for each categorical feature [28].

To explore whether different hyperparameters will have a significant impact on the model performance and to select the best model configurations, we utilized the grid-search method to optimize the hyperparameters for LightGBM using a 2-fold cross-validation experiment on the HepG2 cell line dataset (Supplementary Appendix A).

### 2.5. Training and Evaluation of G4Beacon

In order to obtain a high-performance model that can predict active G4s within a specific cell line robustly, we designed complete training–evaluation approaches for G4Beacon.

As shown in Section 2.2, the ratio of positive and negative samples was highly skewed in each cell line, which would influence the model training if the raw dataset were used as a training set directly. To overcome the data skewness issue, the over-sampling method was applied to the training dataset—that is, the positive samples from the training dataset were over-sampled to the same size as the negative samples (Appendix A).

To evaluate the performance of G4Beacon comprehensively, we designed an evaluation workflow containing two different experiments: a one-cell-line experiment and a cross-cell-line experiment.

In the one-cell-line experiment, the training set and the test set were derived from the same cell line, and the dataset processing steps can be described as follows. First, the raw dataset was divided into two subsets of equal size randomly, and the ratio of positive and negative samples remained the same as in the original dataset. Second, one of the subsets was selected as the training set and the other as the test set. Third, a preprocessing method (over-sampling) was applied to the training set before it was taken as the input of the GBDT classifier for training. Finally, the performance of the trained model was tested on the test set. We applied this evaluation workflow to all the cell line data (K562/HepG2/MCF7) that we collected. The sizes of each training set and test set were displayed (Table 2). We compared the performance divergences of the models with different feature combinations, and they were the sequence-feature-only model, the chromatin-accessibility-feature-only model, and the combined model. Among all combinations, the model achieved the best performance when both sequence features and chromatin features were considered.

The cross-cell-line experiment was applied for further evaluating the robustness of G4Beacon. One of the basic hypotheses of our study is that the active G4s in different cell lines exhibit similar chromatin accessibility and sequence patterns, and therefore the performance of our model should persist when the target samples come from different cell lines. In the cross-cell-line experiment, one cell line dataset was used as the training set and the remaining cell line datasets were used as the test sets. Similar to the one-cell-line experiment, the dataset used for training was preprocessed with over-sampling on positive samples and the datasets for the test remained unchanged. In this study, the HepG2 cell line data were utilized as the training set and the performance was tested on K562 and MCF7 cell line data.

Accuracy, precision, recall, and F1-score were used as the basic criteria of prediction performance. Moreover, ROC and PRC, as well as AUROC and the average precision (AP), were also used to provide a more complete picture of the model’s performance. In both one-cell-line and cross-cell-line experiments, we repeated the process five times with different random seeds and obtained the mean ± standard error for each criterion.
Accuracy = (TP + TN)/(TP + TN + FP + FN)
Precision = TP/(TP + FP)
Recall = TP/(TP + FN)
F1-score = 2 × (Precision × Recall)/(Precision+Recall)
AUROC = Area Under Receiver Operating characteristic Curve
AP = ∑ (Recall_n_ − Recall_n−1_) × Precision_n_

Besides the machine learning evaluation method, we also performed chromatin state analysis to further evaluate our method. Furthermore, we also compared the prediction results of G4Beacon with a recent in vivo G4 prediction tool, DeepG4 [22].

Consistent with the cross-cell-line experiment, we used the full HepG2 dataset balanced with the over-sampling method to train our G4Beacon model. The trained DeepG4 model was obtained from GitHub (https://github.com/raphaelmourad/DeepG4, accessed on 2 November 2022) and directly used for further comparative analysis [22]. The HepG2 and MCF7 datasets were used as test sets, and the input for both methods was kept identical: the human G4-seq entries and ATAC-seq signal track data for each cell line. We verified the active G4 surrounding epigenetic modification states, including H3K4me1, H3K4me2, H3K4me3, H3K9ac, H3K9me3, H3K27ac, and H3K27me3.

## 3. Results

### 3.1. Predicting In Vivo G4s within One Cell Line

We first conducted the one-cell-line experiment using the training set and the test set derived from the same cell line to evaluate the performance of G4Beacon on in vivo G4 prediction. We used the over-sampling method on each training set and employed them to train the model. The performance of the trained models was tested on each testing set using the criteria mentioned in Section 2.5.

We found that G4Beacon exhibited distinctive characteristics in the performance for different in vivo G4 detection techniques (G4 ChIP-seq or G4 CUT&Tag) when discriminating between the positive and negative samples (Figure 3, Table 3, Table 4 and Table 5).

In the G4 ChIP-seq group experiments, we employed data from the K562 cell line and HepG2 cell line. For the K562 cell line, we found that models constructed with different feature combinations exhibited diverse characteristics (Figure 3a,b and Table 3). The sequence-only feature approach had inadequate precision and recall performance in in vivo G4 prediction: F1-score = 0.01, AUROC = 0.93, AP = 0.10. The classifiers using the ATAC-only feature performed much better than the sequence-only model: the F1-score, AUROC, and AP rose to 0.65, 0.99, and 0.67, respectively, in the experiments (Figure 3a,b and Table 3). Finally, we combined the sequence feature and the ATAC feature and found that the results were further improved in comparison with the former models (F1-score = 0.68, AUROC = 1.00, AP = 0.74) (Figure 3a,b and Table 3).

The HepG2 experiment had similar performance to that of the K562 experiment (Figure 3c,d and Table 4). The model using the sequence-only feature approach exhibited poor model performance, both in terms of precision and recall. The ATAC-only model performed well in the K562 cell line, with an F1-score of 0.53, an AUROC of 0.96, and an AP of 0.49 on such an imbalanced test set (Figure 3c,d and Table 4). Moreover, the performance of the seq-ATAC model was also better than that of the other two models (F1-score = 0.54, AUROC = 0.99, AP = 0.58) (Figure 3c,d and Table 4), which was consistent with the results in the K562 cell line.

The performance of G4Beacon in K562/HepG2 experiments exhibited diverse characteristics. First, the sequence-only feature failed to provide enough information for in vivo G4 prediction, which means that the composition of sequences was not sufficient to characterize the formation ability of in vivo G4s. Second, the models using chromatin accessibility features provided excellent prediction accuracy in in vivo G4 prediction. Even on the test set with a skewed positive/negative ratio, G4Beacon could balance the performance in both precision and recall. Finally, adding a sequence feature to our approach improved the performance compared with the chromatin-accessibility-feature-only models. Although the sequence-only models performed poorly, integrating sequence information with chromatin accessibility information did improve the performance in our experiments.

In the G4 CUT&Tag data experiment, we used MCF7 cell line data to construct the training set and the test set. In the MCF7 cell line experiment, G4Beacon with different feature selection approaches performed similarly as on K562 and HepG2 cell line data (Figure 3e,f and Table 5). The sequence-feature-only model showed poor performance, with an F1-score = 0.03, an AUROC = 0.91, and an AP = 0.11 (Figure 3e,f and Table 5). The model trained with the ATAC-only feature also performed well, striking a balance between precision and recall. The combined-feature model also had the best performance among the three feature selection approaches, with an F1-score of 0.52, AUROC of 0.99, and AP of 0.52 (Figure 3e,f and Table 5).

In summary, we validated and confirmed that our model was powerful enough to predict in vivo G4s within both the training set and test set derived from the same cell line.

### 3.2. Cross-Cell-Line Predictions

In this study, we aim to propose a robust in vivo G4 predictor for different human cell lines. Therefore, it is indispensable to test the performance in the situation of a training set and test set coming from different cell lines. We designed a cross-cell-line train–test workflow to evaluate the performance of our model—that is, training on one cell line and testing on other cell lines. Specifically, we conducted a cross-cell-line experiment in which the model was trained on the HepG2 dataset and tested on the other two cell lines, i.e., the K562 cell line and MCF7 cell line. The entire data of each cell line were used in the above experiment. The data for model training were balanced using over-sampling methods, as for the training set in the one-cell-line experiment. As we had confirmed in the former experiments that the combined feature model exhibited the best performance, we then used ATAC and sequence-combined features in the cross-cell-line experiments.

We found that the prediction model trained on HepG2 G4 ChIP-seq data (HepG2 model) in the cross-cell-line experiment had robust performance in predicting in vivo G4s in other cell lines (Figure 4 and Table 6). The HepG2 model provided accurate prediction on the K562 (G4 ChIP-seq data) dataset, with an F1-score of 0.69, AUROC of 1.00, and AP of 0.79 (Figure 4 and Table 6). These results showed the excellent consistency of the data derived from G4 ChIP-seq, which can be learned as in vivo G4 patterns by our approach. The HepG2 model also showed good performance when it was used to predict the in vivo G4s in the MCF7 cell line (G4 CUT&Tag data), giving an F1-score of 0.33, AUROC of 0.98, and AP of 0.40 (Figure 4 and Table 6). Although there was a decrease in prediction performance with lower recall compared to the HepG2train-K562test experiment, our model still maintained acceptable performance, implying that the G4 patterns described by G4 ChIP-seq data are probably stricter than the patterns derived from G4 CUT&Tag data, which is consistent with the feature that G4 CUT&Tag is more sensitive in G4 detection than the G4 ChIP-seq technique [15]. Moreover, we also compared the results between candidate G4-seq entry inputs with canonical G4 motif and non-canonical (two-tetrad format) ones, and found that in vivo G4s with only two tetrads were less predictable (Appendix A). This result confirmed that the two-tetrad G4s were more unstable and easier to be influenced by other potential chromatin environment situations, which was consistent with the prior knowledge [18].

Our G4Beacon approach has been proven to work robustly in the cross-cell-line experiment. Predictions on the K562 dataset, which was divided with the same in vivo G4 detection method data (G4 ChIP-seq), showed prominent precision as well as recall on the whole-genome scale test set. Moreover, G4Beacon can also perform robustly on a dataset (MCF7) constructed with different sequencing techniques. These results demonstrate that G4Beacon is equipped to learn the general ATAC-seq and surrounding sequence patterns of in vivo G4s detected by G4 ChIP-seq/G4 CUT&Tag experiments.

### 3.3. Cell Line Specificity and Histone Modification State of In Vivo G4s Identified by G4Beacon

Previous studies have shown that active G4s are preferentially located in active chromatin regions and are related to some histone modifications [10,12,16]. Inspired by this, we set out to compare the differences in histone modifications around active G4s predicted by G4Beacon and DeepG4.

We characterized the histone modification around in vivo G4s predicted by G4Beacon and DeepG4, which could potentially justify the G4 prediction results as in vivo G4s are preferentially formed in active genomic regions [12,16]. We used the threshold of predicting score 0.5 for both G4Beacon and DeepG4, which means that the resulting G4s are considered to have more than a 50% possibility of being active G4s by the tool. We found that the distribution modes of histone modification signals were varied around the predicted G4s. The G4s predicted by G4Beacon showed higher average signal values and more significant signal peaks than those predicted by DeepG4, regarding the open chromatin histone modifications H3K4me1, H3K4me2, H3K4me3, H3K9ac, and H3K27ac in K562, while the opposite was observed when considering the closed chromatin histone modifications H3K9me3 and K3K27me3 (Figure 5a). Moreover, from the heatmaps, we found that most of the G4s predicted by G4Beacon presented similar signal patterns and intensity, while the DeepG4 results were less consistent (Figure 5a). This was also the case in the MCF7 cell line when we considered the distribution mode of histone modifications around the predicted G4s derived from G4Beacon and DeepG4 (Appendix A). These results coincided with the relationship between G4 ChIP-seq peaks and transcriptional activities [10,16], confirming the rationality of the active G4s predicted by G4Beacon.

Additionally, we used a more stringent threshold (0.95 and 0.99, respectively) for the identification of active G4s and visualized the histone modification states around them, focusing on the most confident results of G4Beacon and DeepG4 (Figure 5b, Appendix A). As a result, the signal values around G4Beacon’s predicted G4s were higher than those around DeepG4’s predicted G4s in those histone modifications that were tightly associated with open chromatin states (Figure 5b, Appendix A).

In summary, we described the histone modification states around the active G4s derived from G4Beacon and DeepG4. The results showed that the G4s predicted by G4Beacon were significantly enriched in open chromatin histone modification regions rather than closed modifications, which implied the characteristics of active G4s.

## 4. Discussion

The development of G4 detection methods, including G4-seq, G4 ChIP-seq, G4P-ChIP-seq, and G4 CUT&Tag [9,10,13,15], has allowed us to profile G-quadruplex structures both in vivo and in vitro. However, it is challenging to perform these methods across multiple cell lines in a limited time and at a limited cost. As a result, a robust computational method is needed to predict the active G4s for different cell lines in a high-throughput manner.

In this research, we focused on implementing the in vivo G4 predictions through a data-driven method. We developed an effective method called G4Beacon, which contained two core modules, namely the complete feature-construct module and the predictor module. In the feature-construct module, we used chromatin accessibility data, i.e., ATAC-seq data and the sequence data, as input to describe a candidate G4 sample. The predictor module was built based on a strong ensemble learning method, GBDT, and we used one of the widespread implements: LightGBM [28].

Different from most of the existing G4 prediction tools that took only sequence data as input and predict non-cell-type-specific G4s, our model utilized the chromatin accessibility profiles, which reflect the intracellular environments of cells within G4 folding regions, as well as the surrounding sequences, which contain information about regulatory elements related to G4s, e.g., transcription factor binding sites. Due to the importance of in vivo G4 prediction, another deep learning algorithm has recently been proposed to characterize and predict in vivo G4s based on chromatin accessibility and sequence patterns, namely DeepG4 [22]. Although the DeepG4 tool achieved fair AUROC performance on data from different cell lines, the AUPRC results remained to be greatly improved, which reflected the potential high-false-positive-ratio problem. To overcome this problem, we employed a different data preprocessing strategy. Our method used a wider surrounding region of 2000 bp length for feature construction, which helped the model to acquire more cellular environment and surrounding sequence pattern information for each candidate G4. The results showed that although our model was trained on a specific cell line, its superior performance persisted in other cell lines, meaning that it can be transplanted to those cell lines whose G4s are not depicted.

To verify the performance of G4Beacon, we profiled both machine learning evaluation experiments and bioinformatics analysis of histone modifications around the predicted active G4s. The machine learning evaluation experiments, including one-cell-line and cross-cell-line experiments, showed that G4Beacon not only performed well on data from one cell line but was also robust in cross-cell-line predictions (Figure 3 and Figure 4). Moreover, the results of the histone modification analysis also supported the notion that the active G4s identified by G4Beacon had reliable cell line specificity and correlated with some important histone modifications (Figure 5). Therefore, our method was relatively reliable in G4 prediction scenarios.

However, the features that we considered in constructing the G4Beacon model are still inadequate to completely describe the cellular environment involved in G4 folding, which is a limitation of this model. Although the sequence patterns can be different between active and inactive G4s, the main differences should arise from the cell line environment where they are harbored—that is, the chromatin accessibility feature used in our research. With the developments in researching the environmental factors that can influence the formation of in vivo G4s, more useful information can be taken as input features. Even so, the framework of our G4Beacon can be generalized to support any type of sequential environment data. It is conceivable that G4Beacon can easily be employed to train on new expressive features, and thereby we can acquire predictors with better performance.

In conclusion, G4Beacon is a very accurate, stable, and concise tool that can be used for the prediction of in vivo G4s.

## 5. Conclusions

In this research, we proposed G4Beacon, a machine-learning-based in vivo G4 prediction method that utilized both the sequence and chromatin accessibility information to depict and identify the cell-type-specific active G4s. We tested G4Beacon in a one-cell-line experiment and a cross-cell-line experiment, which showed that our model was considerably accurate and robust in whole-genome in vivo G4 prediction. In summary, our method can quickly and extensively describe the landscape of in vivo G4s in existing cell lines and can facilitate genome-wide G4 studies and the discovery of their potential new biological functions.

## Figures and Tables

**Figure 1 biomolecules-13-00292-f001:**
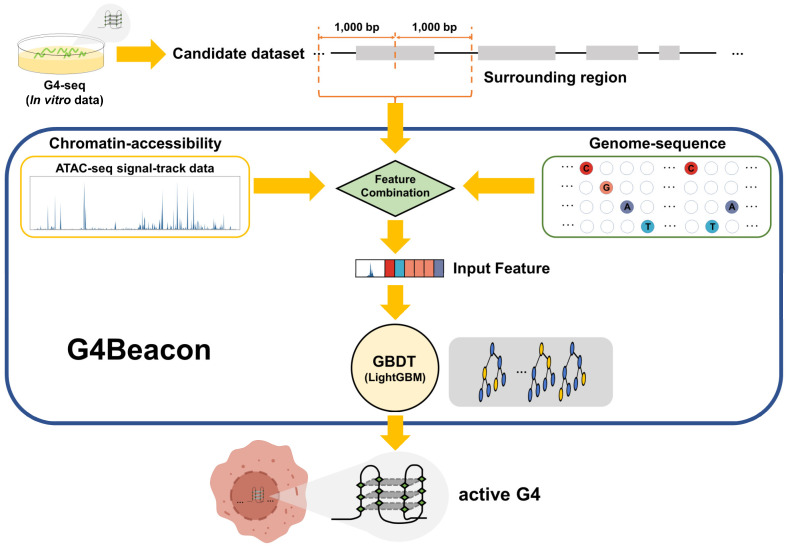
Overview of G4Beacon workflow. G4-seq data are employed as the candidate active G4 dataset. For each candidate sample (G4-seq entry), G4Beacon extends it into a region of 2000 bp length (surrounding region) and utilizes both ATAC-seq signal-track data and sequence data for feature selection and construction. The ATAC-seq feature is transformed into a float vector using the sliding-window method and the sequence feature is encoded using the ordinal categorical encoding method. All of these features are taken by our trained-GBDT model (implemented using LightGBM) as input for predicting whether the G4 will fold or not in the specific cellular environment.

**Figure 2 biomolecules-13-00292-f002:**
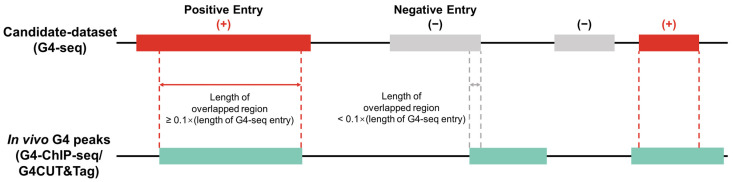
Determination of positive and negative samples. In the training and evaluation workflow, in vitro G4 data (G4-seq) are used as the candidate dataset. In vivo G4 experimental data (G4 ChIP-seq or G4 CUT&Tag) are used to split positive (active G4s) and negative (inactive G4s) samples. More specifically, a candidate G4 is tagged as a positive sample if it overlaps with an in vivo G4 peak and the length of the overlapped region is greater than 10% of this candidate entry’s length.

**Figure 3 biomolecules-13-00292-f003:**
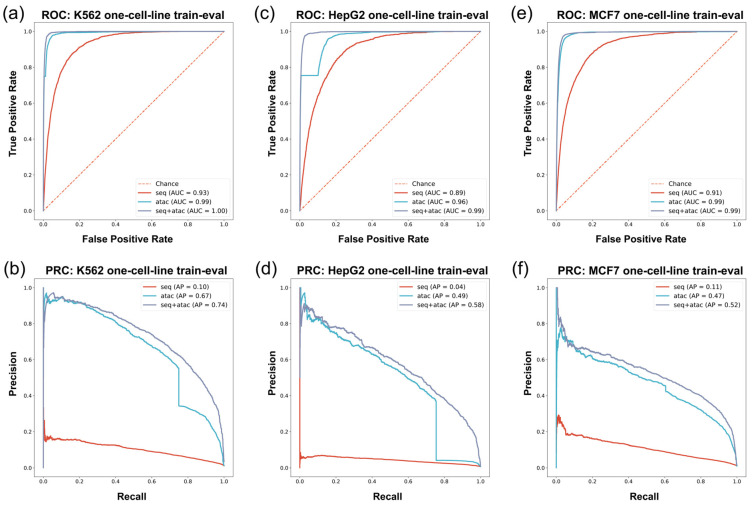
ROC/PRC of in vivo G4 prediction with training set and test set derived from same cell line. The vertical axis and horizontal axis are True Positive Rate = TP/(TP+FN) and False Positive Rate = FP/(FP + TN) for ROC, precision, and recall for PRC. Since in vivo G4 prediction is an imbalanced data problem where the number of positive samples is overwhelmingly more than that of negative samples, the Area Under PRC (AP) can reflect the performance of models more effectively. (**a**,**b**) The ROC/PRC of K562 cell line. (**c**,**d**) The ROC/PRC of HepG2 cell line. (**e**,**f**) The ROC/PRC of MCF7 cell line.

**Figure 4 biomolecules-13-00292-f004:**
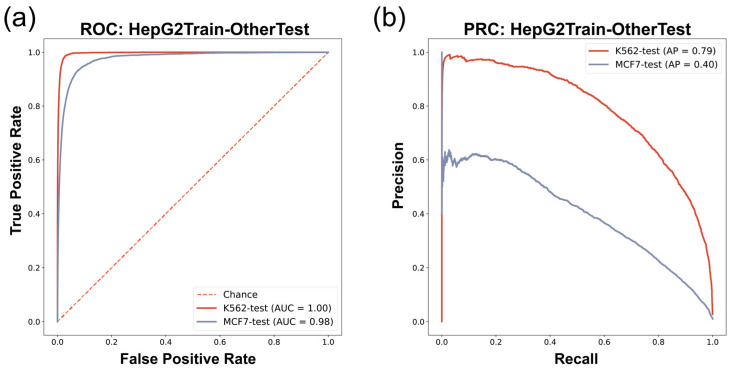
Cross-cell-line evaluation ROC/PRC results. (**a**) ROC of HepG2-dataset-trained model predicting in vivo G4s of K562/MCF7 cell lines. (**b**) PRC of HepG2-dataset-trained model predicting in vivo G4s of K562/MCF7 cell lines.

**Figure 5 biomolecules-13-00292-f005:**
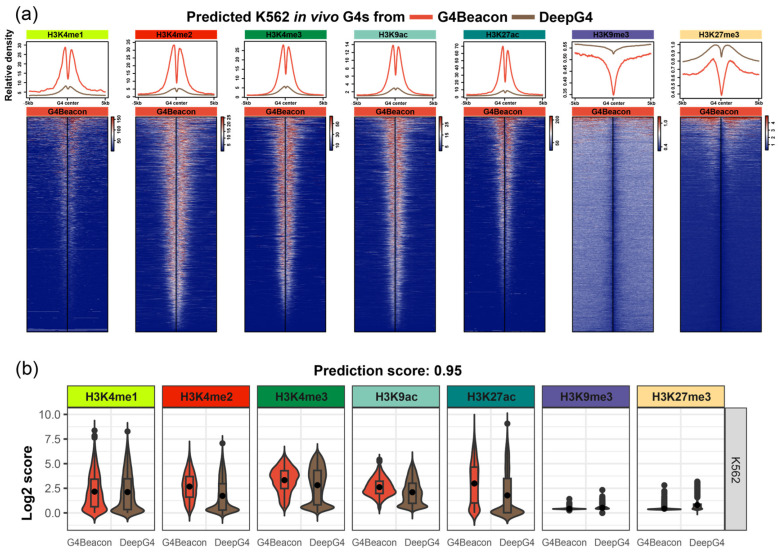
Results of the histone modification state analysis on G4Beacon and DeepG4 prediction results. (**a**) The line charts and heatmaps of histone modification states around K562 active G4s derived from G4Beacon and DeepG4, with the probability threshold of 0.5. (**b**) The boxplots of histone modification state scores of K562 active G4s derived from G4Beacon and DeepG4, with the probability threshold of 0.95.

**Table 1 biomolecules-13-00292-t001:** Positive/negative samples of different cell lines.

Cell Line	Positive Samples	Negative Samples
K562	3716	430,372
HepG2	2491	431,597
MCF7	4272	429,816

**Table 2 biomolecules-13-00292-t002:** Training set and test set sizes of experiments for each cell line.

Cell Line	Training Set Positive/Negative	Test Set Positive/Negative
K562	215,186/215,186	1858/215,186
HepG2	215,798/215,798	1246/215,799
MCF7	214,908/214,908	2136/214,908

**Table 3 biomolecules-13-00292-t003:** K562 cell line training–evaluation results.

	Accuracy	Precision	Recall	F1-Score	AUROC	AP
seq	0.99 ± 0.00	0.13 ± 0.02	0.01 ± 0.00	0.01 ± 0.00	0.93 ± 0.00	0.10 ± 0.00
ATAC	0.99 ± 0.00	0.66+0.01	0.65+0.00	0.65+0.00	0.99+0.00	0.67+0.01
ATAC+seq	0.99 ± 0.00	0.70+0.01	0.66 ± 0.00	0.68 ± 0.00	1.00 ± 0.00	0.74+0.01

Each criterion is expressed as mean ± standard error. seq: sequence-only; ATAC: ATAC-only; seq+ATAC: sequence–ATAC combined.

**Table 4 biomolecules-13-00292-t004:** HepG2 cell line training–evaluation results.

	Accuracy	Precision	Recall	F1-Score	AUROC	AP
seq	0.99 ± 0.00	NaN	NaN	NaN	0.89 ± 0.00	0.04 ± 0.00
ATAC	0.99 ± 0.00	0.57 ± 0.01	0.50 ± 0.01	0.53 ± 0.00	0.98 ± 0.01	0.49 ± 0.02
ATAC+seq	0.99 ± 0.00	0.63 ± 0.00	0.48 ± 0.00	0.54 ± 0.00	0.99 ± 0.00	0.58 ± 0.00

**Table 5 biomolecules-13-00292-t005:** MCF7 cell line training–evaluation results.

	Accuracy	Precision	Recall	F1-Score	AUROC	AP
seq	0.99 ± 0.00	0.22 ± 0.02	0.02 ± 0.00	0.03 ± 0.00	0.91 ± 0.00	0.11 ± 0.00
ATAC	0.99 ± 0.00	0.48 ± 0.00	0.53 ± 0.01	0.51 ± 0.00	0.99 ± 0.00	0.47 ± 0.00
ATAC+seq	0.99 ± 0.00	0.55 ± 0.00	0.48 ± 0.01	0.52 ± 0.00	0.99 ± 0.00	0.52 ± 0.00

**Table 6 biomolecules-13-00292-t006:** Cross-cell-line prediction results (HepG2-trained model).

	Accuracy	Precision	Recall	F1-Score	AUROC	AP
Test on K562	0.99 ± 0.00	0.79 ± 0.00	0.61 ± 0.00	0.69 ± 0.00	1.00 ± 0.00	0.79 ± 0.00
Test on MCF7	0.99 ± 0.00	0.59 ± 0.00	0.23 ± 0.00	0.33 ± 0.00	0.98 ± 0.00	0.40 ± 0.00

## Data Availability

The human G4-seq data, the G4 ChIP-seq data for K562 and HepG2, and the G4 CUT&Tag data for MCF cell lines were downloaded from the Gene Expression Omnibus (GEO) database under the accession numbers of GSE110582 [24], GSE107690 [12], GSE145090 [31], and GSE181373 [15], respectively. All the ATAC-seq data were obtained from the Encyclopedia of DNA Elements (ENCODE) portal with the identifiers of ENCFF357GNC, ENCFF262URW, and ENCFF976UNK for the cell lines of K562, HepG2, and MCF7, respectively. The histone modification state data were also obtained from the ENCODE database with the identifiers of ENCFF783QIW (K562-H3K4me1), ENCFF060FHP (K562-H3K4me2), ENCFF370CHI (K562-H3K4me3), ENCFF541RTV (K562-H3K9ac), ENCFF422BDQ (K562-H3K9me3), ENCFF465GBD (K562-H3K27ac), ENCFF796REQ (K562-H3K27me3), ENCFF199WYG (MCF7-H3K4me1), ENCFF024VOG (MCF7-H3K4me2), ENCFF615NAU (MCF7-H3K4me3), ENCFF917IBG (MCF7-H3K9ac), ENCFF642EON (MCF7-H3K9me3), ENCFF246QSB (MCF7-H3K27ac), and ENCFF216VVY (MCF7-H3K27me3). All code is available at the GitHub repo: https://github.com/Bocabbage/G4Beacon, accessed on 2 November 2022.

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
