# Peer review of "G4Beacon: An In Vivo G4 Prediction Method Using Chromatin and Sequence Information"

_biomolecules, 2023, doi:10.3390/biom13020292_

Round 1
Reviewer 1 Report
This article by Zhang et al. deals with a machine-learning-based in vivo G-quadruplex prediction method, named G4Beacon, exploiting both the sequence and chromatin accessibility information. Authors provided the complete workflow of the method and the accurate description of its training before performing the prediction. As the only revision, I would suggest to add the errors, where possible, for the data in Tables 3-6.
Reviewer 2 Report
In this manuscript, Zhuofan Zhang and his co-workers have described a reliable method for the prediction of in vivo G4s of various cell lines. I consider these results potentially interesting; this work is sufficiently exhaustive and well-written.
However, some more information should be introduced on the type of cells considered for prediction model in vivo (for example, the type of cancer in which each cell line is involved), and on the reason for the choice of these specific cell lines.
In my opinion, once it has been subjected to these minor revisions, this work will be suitable for publication in " Biomolecules".
Reviewer 3 Report
Overall, this manuscript demonstrates G4Beacon, which is a machine-learning based method that can predict in vivo G4 formation in specific cell lines. As a result, the model gives powerful predicting results of in vivo G4s. This is interesting work that is worth of publication in Biosensors. However, the following concerns should be addressed before publication.
1. As a computational method to predict G4 sequences, QGRS mapper has been widely used. Although it can not predict non-canonical G4s, it still plays an important role in G4 study field. Please add the according information in the introduction part.
2. In Figure 2, the positive entry is considered when more than 10% overlap between G4-seq and G4 ChIP-seq/G4 CUT&Tag is observed. Is there any rational for the 10% threshold? If there is any reference, please cite. If not, please add explanation in the corresponding paragraph.
3. According to the G4-seq data, there are some non-canonical G4s forming in cells. Compared to the canonical ones, G4s with only two tetrads seems more unstable and easier to get influenced by histone modification state. Did the authors observe any significant difference in prediction scores between two-tetrad G4s and canonical ones (3Gs, 4Gs, and 5Gs)?
